# POSCAbilities: The Application of the Prion Organotypic Slice Culture Assay to Neurodegenerative Disease Research

**DOI:** 10.3390/biom10071079

**Published:** 2020-07-20

**Authors:** Hailey Pineau, Valerie Sim

**Affiliations:** 1Centre for Prions and Protein Folding Diseases, University of Alberta, 204 Brain and Aging Research Building, 8710-112 St, Edmonton, AB T6G 2M8, Canada; pineau@ualberta.ca; 2Neuroscience and Mental Health Institute, University of Alberta, 2-132 Li Ka Shing, Edmonton, AB T6G 2E1, Canada; 3Division of Neurology, Department of Medicine, Faculty of Medicine & Dentistry, University of Alberta, Clinical Sciences Building, 8440-112 St, Edmonton, AB T6G 2B7, Canada

**Keywords:** prion, organotypic slice culture, neurodegenerative disease, protein aggregation, protein misfolding, Alzheimer’s disease, Parkinson’s disease, amyotrophic lateral sclerosis, Huntington’s disease

## Abstract

Prion diseases are fatal, transmissible neurodegenerative disorders whose pathogenesis is driven by the misfolding, self-templating and cell-to-cell spread of the prion protein. Other neurodegenerative diseases such as Alzheimer’s disease, Parkinson’s disease, amyotrophic lateral sclerosis and Huntington’s disease, share some of these prion-like features, with different aggregation-prone proteins. Consequently, researchers have begun to apply prion-specific techniques, like the prion organotypic slice culture assay (POSCA), to these disorders. In this review we explore the ways in which the prion phenomenon has been used in organotypic cultures to study neurodegenerative diseases from the perspective of protein aggregation and spreading, strain propagation, the role of glia in pathogenesis, and efficacy of drug treatments. We also present an overview of the advantages and disadvantages of this culture system compared to in vivo and in vitro models and provide suggestions for new directions.

## 1. Introduction

Prions are infectious proteins that cause rapidly progressive, fatal neurodegenerative disorders [1]. Such diseases include Creutzfeldt Jakob disease (CJD) in humans, bovine spongiform encephalopathy (BSE) in cows, scrapie in sheep, and chronic wasting disease (CWD) in cervids [2]. Prion diseases are caused by the misfolding of the prion protein (PrP) from its non-toxic cellular state (PrP^C^) to PrP^Sc^, which exists as soluble oligomers and beta-sheet-rich fibrils and aggregates. Because PrP^Sc^ self-templates the misfolding of PrP^C^, PrP^Sc^ is infectious and spreads from cell-to-cell and host-to-host [1,2]. In humans, prion disease is very rare with an incidence of about 1–2 people per million per year [3]. However, mounting evidence suggests that the proteins implicated in other neurodegenerative diseases, including amyloid beta (Aβ) and tau in Alzheimer’s disease (AD), alpha synuclein (α-syn) in Parkinson’s disease (PD), TAR DNA binding protein (TDP-43) and superoxide dismutase (SOD1) in amyotrophic lateral sclerosis (ALS), and huntingtin (Htt) in Huntington’s disease (HD) spread via prion-like mechanisms [4,5,6,7,8,9,10,11,12].

Another prion-specific concept that has proven applicable to other neurodegenerative diseases is the concept of strains. In prion disease, strains are clinically and pathologically distinct phenotypes thought to arise from different conformations of PrP^Sc^. Each prion strain is associated with a distinct incubation period, rate of disease progression, and symptom profile. Pathologically, strains target different brain regions and have unique PrP^Sc^ deposition patterns [13]. Additionally, strains can be characterized biochemically by their variable conformational stability, sensitivity to protease digestion, aggregation kinetics, and glycosylation pattern seen in western blot [14]. Increasingly, researchers have demonstrated that strains account for at least some of the heterogeneity observed in other neurodegenerative diseases, including AD, PD, and ALS [15,16,17,18,19].

The prion-like spreading observed in AD, PD, ALS, frontotemporal dementia (FTD), and HD has allowed for the expansion of prion-specific techniques to study and model these more prevalent diseases. One such technique is the prion organotypic slice culture assay (POSCA), in which organotypic slice cultures are exposed to infectious prions and recapitulate pathology, including strain differences, as seen in vivo [20]. This review will discuss how organotypic slice culture has been used to increase our understanding of the prion-like nature of other neurodegenerative diseases, specifically seeded aggregation and cell-to-cell spread, strain propagation, the role of glia in pathogenesis, and efficacy of drug treatments.

## 2. Prion Organotypic Slice Culture Assay (POSCA)

### 2.1. The Origin and Evolution of Prion Organotypic Slice Culture Assay

The POSCA was developed in 2008 by Falsig and Aguzzi. They demonstrated that cerebellar organotypic slice cultures from wildtype mice can be infected with multiple prion strains and cultured for up to three months. Proteinase K-resistant PrP^Sc^ from the slices is detectable using western blot by 21 days post-infection (dpi) and approaches levels comparable to that seen at terminal in vivo infection by 35 dpi [20]. Real-time quaking-induced conversion (RT-QuIC), a prion amplification technique used to diagnose disease, can detect PrP^Sc^ in POSCA as early as seven dpi [21]. Thus, PrP^Sc^ accumulation occurs on an accelerated time scale in POSCA compared to in vivo infection. As expected, no PrP^Sc^ is observed at any time point in Prnp-knockout slices [20]. The POSCA can also be used as a diagnostic tool for detecting low titres of prion infectivity. For example, cerebellar slice cultures are capable of propagating infection from white blood cells isolated from presymptomatic scrapie-infected sheep [22]. Unlike the scrapie cell assay in which prion infectivity is assessed in neuroblastoma (N2a) or catecholaminergic-derived (CAD) cells, POSCA allows for the study of prion infection in a complex environment in which the in vivo cytoarchitecture is largely preserved. Moreover, organotypic slice cultures from wildtype mice can be infected with many strains, including the mouse-adapted scrapie strain ME7 and the BSE-derived strain 301C, which cannot be propagated in N2a or CAD cells [20]. The widespread availability of transgenic mice further expands the types of prion diseases that can be studied in POSCA. Kondru et al. recently found that cerebellar slices from Tg12 mice, which express elk Prnp, can be infected with and propagate CWD prions [23]. We have also adapted the POSCA cerebellar culture system to coronally sliced whole brain (Figure 1), successfully propagating scrapie strains RML, 22L, and ME7 (Figure 2). This allows examination of patterns of PrP deposition in different brain regions with different strains. Moreover, dividing the sections sagittally allows the use of one hemisphere as an uninfected or untreated control from the same mouse and matched brain region. Alternately, different strains can be applied to matched slices from the same mouse to compare strain propagation. Beyond immunoblotting slice homogenates for PK-resistant PrP, there are a variety of ways to image PrP deposition in the slices, including histoblot [24], heat mapping of total PrP levels (which includes PrP^C^ and PrP^Sc^, Figure 2), or confocal imaging of aggregates of PrP, labelled with antibodies after epitope retrieval [25] (Figure 1). While an immunoblot can demonstrate presence of PrP^Sc^ within a slice homogenate, imaging the intact slice allows much more information to be gleaned about the distribution of PrP and pathology.

#### 2.2. Pathology in Prion Organotypic Slice Culture Assay

Many pathological features of in vivo prion infection are recapitulated in POSCA. By 42 dpi in RML-infected cerebellar slices, there is significant loss of NeuN and synaptophysin in the granule cell layer compared to non-infected controls, and intense PrP-immunostaining is observed in this region [25,26]. Astrogliosis, increased microglia, and spongiform vacuolation are also observed [24,26], and loss of dendritic spines and changes in spine morphology have been reported by 63 dpi [27]. Interestingly, histoblots show that organotypic slice cultures exhibit strain differences typical of in vivo infection. Diffuse prion deposition is seen in RML strain-infected slices, while the 22L strain induces dense, multifocal plaques, and the 139A strain leads to patchy PrP^Sc^ deposition in the white matter [24].

The POSCA has also been used to better understand downstream pathological mechanisms of prion infection. Specifically, Harischandra et al. investigated whether the protein kinase C Cδ (PKCδ), a pro-apoptotic kinase, is involved in prion-induced neuronal death. They infected cerebellar slices from C57BL/6 mice with RML prions. After two weeks, they found more intense PKCδ immunoreactivity in the Purkinje cells of the molecular layer compared to non-infected control slices. They also observed both an increase in PKCδ mRNA and enhanced proteolytic activation of PKCδ in infected slices, suggesting an important role for this kinase in prion-induced cell death [28]. Herrmann et al. found that exposure of cerebellar Tga20 organotypic cultures to RML prions led to significant reactive oxygen species (ROS) production and elevated levels of phosphorylated PERK, eIF2α, and ATF4, which are all markers of endoplasmic reticulum stress. Treatment of slice cultures with ROS scavengers such as ascorbate and isoascorbate was protective, though these compounds did not affect the accumulation of PrP^Sc^ [29].

### 2.3. Manipulation of Glia in Prion Organotypic Slice Culture Assay

The POSCA is particularly amenable to the study of microglia as they can be suppressed or removed by various methods. Zhu et al. prepared cerebellar slice cultures from tga20/CD11b-thymidine kinase of herpes simplex virus (HSV TK) mice, in which microglia can be conditionally ablated through addition of granciclovir. After infection with RML prions, microglia-depleted slices had significantly fewer granular neurons compared to infected slices in which microglia had been retained [30]. A previous experiment from the same group also showed up to 5x greater PrP^Sc^ deposition in microglia-depleted slices [26]. It is also possible to add glial cells to POSCA to study mechanisms of prion spread. Specifically, Hofmann et al. used POSCA to investigate spreading of the N-terminal and middle domain of Sup35 (NM), a yeast prion protein. They infected astrocytes expressing soluble NM with recombinant NM fibrils, which were then added to organotypic mouse hippocampal slices that expressed soluble NM. After 12 days of culture, they found intracellular NM aggregates within the slice [31].

### 2.4. Prion Organotypic Slice Culture Assay as a Tool for Drug Screening

The POSCA has proven useful for testing anti-prion drugs. Cortez et al. found a dose-dependent neuroprotective effect of bile acids tauroursodeoxycholic acid (TUDCA) and ursodeoxycholic acid (UDCA) in RML-infected cerebellar slices from Tga20 mice. Treatment with either compound at seven or 14 dpi led to a significantly lower level of PrP^Sc^ without affecting levels of PrP^C^. Treatment starting at 21 days was not as effective [32]. Margalith et al. tested poly(thiophene-3-acetic acid) (PTAA), a luminescent conjugated polythiophene (LCPs), for its ability to slow prion replication in RML-infected Tga20 cerebellar slice cultures. Interestingly, they observed that PTAA dose-dependently reduced accumulation of PrP^Sc^. However, the resulting PrP^Sc^ was actually more resistant to proteolysis [33]. Moreover, Wolf et al. observed that the sulfated polysaccharide DS-500 was effective at reducing PrP^Sc^ accumulation and astrocytosis in 22L-infected cerebellar slices compared to control [25]. In another experiment, inhibitors of the mGluR5 receptor, including methyl-6-(phenylethynyl)-pyridine (MPEP), were found to be neuroprotective in RML-infected cerebellar and hippocampal slices. Additionally, knockout of mGluR5 receptors in hippocampal slices reduced cell death in infected slices, confirming the salience of this target for drug intervention [34]. In the CWD-POSCA experiments performed by Kondru et al., seeding activity was significantly reduced by congo red, while quinacrine and astemizole were only mildly effective or ineffective, respectively [23].

## 3. Slice Culture to Study Prion-Like Mechanisms in Alzheimer’s Disease

### 3.1. Slice Culture from Alzheimer’s disease Transgenic Mouse Models

Many groups have made use of organotypic slice culture for the study of AD. Harwell and Coleman cultured cortico-hippocampal slices from neonatal TgCRND8 mice, which overexpress human APP with both the Swedish and Indiana mutations. These slices could be cultured for at least nine weeks, and many features of AD were recapitulated including loss of the presynaptic protein synaptophysin. Moreover, intracellular Aβ accumulated in calbindin-positive axonal swellings withing the CA1 regions of the hippocampus between two and five weeks. Although Aβ aggregates are known to develop in vivo in CRND8 mice by nine weeks, the authors did not observe any thioflavin-S (Th-S)-positive Aβ aggregates in slice culture by that time [35]. Novotny et al. had similar findings. They cultured hippocampal slices from neonatal APP23 mice, which overexpress human APP with the Swedish double mutation, and APPPS1 mice, which overexpress APP with the Swedish mutation and express human mutant presenilin-1 (PS1). Despite Aβ plaques being present in vivo by 8 weeks, no Aβ aggregates were detected in culture by 10 weeks [36]. The lack of aggregates in ex vivo cultures may be due to much of the Aβ being released into the medium, which is typically changed every 2–3 days. Novotny et al. noted that the slice tissue contained ~0.1% of the Aβ found in the medium [36]. Similarly, Croft et al. observed no sign of Aβ aggregates in hippocampal slice culture from 3xTg-AD mice, which express mutant human PS1, human APP with the Swedish mutation, and human mutant P301L tau. However, they found that these slices had accumulated similar levels of Aβ42 and phosphorylated tau by 28 days in culture as levels at 12 months in vivo, suggesting that these proteins accumulate on an accelerated time scale ex vivo [37,38].

### 3.2. Application of Exogenous Amyloid Beta to Promote Aggregate Formation

Although Aβ aggregates have not been observed in unmodified slice cultures from AD transgenic mice, several groups have tried inducing prion-like spreading and aggregate formation by adding exogenous Aβ seeds to the cultures. Novotny et al. supplemented hippocampal cultures with synthetic Aβ40 or Aβ42 in the medium throughout the culture period and pipetted brain homogenate from aged APPPS1 or APP23 mice on top of the slices once at the beginning of the culture period. After only 1 week, Aβ aggregates were detected in the slices. This was observed not only in transgenic APPPS1 and APP23 slices, but also wildtype and APP-null slices. However, aggregates were only observed when both forms of exogenous Aβ were applied. Several AD-pathological features were observed in these plaque-containing slices, including the presence of phosphorylated tau in dystrophic neurites and a 50% decrease in dendritic spines compared to untreated slices. Interestingly, the type of aggregates that formed depended on the source of the brain homogenate; homogenate from APP23 mice elicited fewer large aggregates, and homogenate from APPPS1 mice induced more numerous, small, compact aggregates, which mirrors the morphotypes of plaques seen in vivo in these mouse models [36]. This suggests that slice culture may provide a useful medium for investigating AD strain differences.

### 3.3. Manipulation of Microglia

It has become increasingly apparent that microglia play a significant role in the clearance of Aβ, and it has been observed that Aβ aggregates added to rat hippocampal slice culture are quickly consumed by microglia [39]. Therefore, researchers have also facilitated prion-like spreading of Aβ in slice culture by applying the drug clodronate to eliminate microglia. Hellwig et al. treated cultured hippocampal slices from neonatal WT mice with synthetic Aβ42. Only when slices were treated with clodronate to eliminate microglia did they develop ThS-positive Aβ aggregates after two weeks [40]. Similar results were observed by Richter et al., who also found that treatment of slice cultures with cytochalasin D, a phagocytosis inhibitor, led to the formation of Aβ aggregates when exogenous Aβ42 was applied [41].

### 3.4. Drug Screening

Several groups have made use of slice culture treated with exogenous Aβ to screen for potential AD drugs. Specifically, many researchers have tested drugs that target ROS. Alberdi et al. found that the antioxidants mangiferin and morin were effective at reducing ROS in cortical rat brain cultures exposed to 5 µM of Aβ oligomers [42]. Similarly, Arbo et al. observed that 4-cholorodiazepam was neuroprotective in such Aβ-treated rat brain cultures and led to increased superoxide dismutase expression [43]. In addition, Campolo et al. found that dimethyl fumerate was effective for reducing tau phosphorylation and ROS in CD1 mouse brain slices treated with exogenous Aβ [44]. Others have tested drugs for their ability to inhibit Aβ aggregation in slice culture. In slice cultures of rat hippocampus treated with Aβ42, curcumin was able to prevent cell death and a decrease in phosphorylation of synapsin I and CaMKII, which is required for synaptic transmission [45]. Additionally, Barucker et al. tested an Aβ42-interacting peptide (AIP) in Aβ42-treated hippocampal mouse brain slices. They had previously shown that this peptide prevents the fibrillization of Aβ oligomers. In slice culture, they found that AIP prevented a loss of dendritic spines and a decrease in long-term potentiation (LTP) [46].

Because no transgenic AD mouse model fully recapitulates all aspects of AD, it would be useful to develop organotypic cultures from human brain tissue. Although not as readily available, one source of human brain tissue has been from patients undergoing medial temporal lobectomy for treatment of refractory epilepsy. One group was able to culture 200 µm slices from such cortical tissue and found that it remained electrophysiologically active and viable for at least four days ex vivo. However, by nine days of culture, there was a 50% decrease in cell viability. Although the culture was short-lived, they found that application of Aβ oligomers led to elevated levels of phosphorylated tau [47]. This system is arguably more biologically relevant for the screening of AD drugs compared to neonatal mouse brain slices. Nevertheless, more work needs to be done to optimize and prolong the health of these slices so that they may be used in longer term studies.

### 3.5. Tau Aggregation in Slice Culture

The past few years has also seen several studies in which organotypic slice culture was used to study tau pathology. Messing et al. cultured hippocampal slices from transgenic mice expressing 4-repeat tau with the frontotemporal dementia mutation ΔK280. Several hallmarks of tauopathy were observed in culture, including hyperphosphorylation and mislocalization of tau from axons to the soma and dendritic compartments by 5 days ex vivo. Furthermore, they observed a progressive loss of dendritic spines and cell death and the formation of tau aggregates by 3–4 weeks. Addition of the rhodamine-based tau aggregation inhibitor bb14 significantly prevented this pathology [48]. Stancu et al. cultured hippocampal slices from transgenic mice with the P301S tau mutation and treated them with tau fibrils. After 10 days, there was increased tau immunostaining in the hippocampus compared to wildtype-seeded slices and non-seeded transgenic slices. Moreover, EM analysis revealed the presence of tau fibrils in the seeded transgenic cultures, but not in non-seeded transgenic or seeded wildtype cultures. The seeded transgenic cultures had also accumulated more tau oligomers than the other slices [49].

### 3.6. Tau Spreading

Other groups have investigated the modes by which tau is taken up by and spreads through organotypic cultures. For instance, it has been demonstrated that tau-containing exosomes in media are more efficiently taken up by cultures compared to free non-exosomal tau. Moreover, it was found that most exosomal tau was taken up by neurons and microglia, and not astrocytes [50]. Suttkus et al. were interested in how the perineuronal net (PN) affects movement of tau in organotypic cultures. They cultured brain slices from wildtype mice as well as knockouts for various components of the PN, including aggrecan, HAPLN1, and tenascin-R. They observed that exogenously-applied tau oligomers were able to move farther within slices that were missing components of the PN and that neurons not ensheathed by a PN more frequently showed tau internalization [51]. Thus, the PN restricts both movement and internalization of tau in slice culture.

## 4. Slice Culture to Study Prion-Like Mechanisms in Parkinson’s Disease

### 4.1. Induction of Rodent α-Syn Aggregation and Spreading

Several researchers have made use of organotypic slice culture to study Parkinson’s disease and other synucleinopathies. Much evidence suggests that a loss of dopaminergic neurons in the substantia nigra is a trigger for the aggregation of α-syn and formation of Lewy bodies (LBs) [52]. Therefore, Cavaliere et al. created an ex vivo model of PD in which nigrostriatal projections were mechanically severed in sagittal brain slices from Sprague-Dawley rats. After 3–5 days in culture, there was a 60% decrease in dopaminergic neurons, and α-syn aggregates were observed in the caudoputamen. Similar results were also achieved when slices were treated with 6-OHDA, which selectively kills dopaminergic neurons [53]. Other groups have generated ex vivo PD models by exposing organotypic cultures to toxins that are known to cause parkinsonism in vivo. For instance, exposure to the heavy metal manganese causes a form of Parkinson’s disease known as manganism [54]. Xu et al. performed a series of experiments in which brain slices from wistar rats were exposed to up to 400 µM of manganese for 24 h. There was significantly greater ROS generation and cell death in treated slices and a concentration-dependent increase in both α-syn mRNA and α-syn oligomers in the basal ganglia [55,56,57]. Another toxin known to cause parkinsonism is the pesticide rotenone, and it is widely theorized that PD induced by exposure to such toxic compounds may originate from peripheral regions including the intestinal tract or nasal mucosa and then spread to the brain [58,59,60]. Therefore, Sharrad et al. cultured organotypic slices of guinea pig ileum, which were exposed to 10 µM of rotenone. α-syn fibrils were detected in the myenteric plexus after one day of exposure, the tertiary plexus after two days, and the circular muscle plexus after four days, supporting a prion-like spreading of α-syn [61].

### 4.2. Induction of Human α-Syn Aggregation

It has also been possible to study PD in organotypic cultures that express human α-syn. Croft et al. created an organotypic model of PD by transfecting hippocampal mouse brain slices with recombinant adeno-associated virus (rAAV) that expressed either human wildtype or A53T mutant α-syn. Expression of α-syn in the slices remained stable for the entire 90 days of culture, and expression could be achieved in all cells using the hCBA promotor, neurons with the CamKII or MAPT promotors, astrocytes with the GFAP promotor, or microglia with the CD68 promotor. By 28 days in culture, α-syn LB-like inclusions and Lewy neurites were observed in the slices that expressed either WT or A53T α-syn [62].

### 4.3. Application of Exogenous α-Syn to Promote Aggregation

Researchers have also added exogenous α-syn to organotypic culture to induce seeding and prion-like spread of α-syn fibrils. One group cultured hippocampal slices from rats that were treated with either monomeric α-syn, fibrillar α-syn, or both. It was found that α-syn by itself was not toxic to slice cultures, though fibrillar α-syn induced moderate cell death. However, adding both monomeric and fibrillar α-syn led to a much greater loss in cell vitality. The researchers thus concluded that it was the fibrillization of α-syn, requiring both preformed fibrils and monomers, that was toxic to the slices. This was further confirmed upon addition of talcapone, which inhibits fibrillization, or β-synuclein instead of α-syn monomers, both of which prevented cell death [63]. In another experiment performed by Elfarrash et al., the dentate gyrus of organotypic hippocampal mouse brain slices was microinjected with α-syn fibrils. a-syn aggregates first appeared in the dentate gyrus after three days. After 5–7 days, α-syn inclusions were also detected in CA3 and CA1. Interestingly, no fibrils were detected when monomeric α-syn was injected or when slices were prepared from an SNCA knockout mouse. When α-syn fibrils were microinjected into CA1, α-syn aggregates were observed in CA1 after 14 days, but no retrograde spreading occurred to CA3 or the dentate gyrus [64].

### 4.4. The Role of Cell Type in α-Syn Spreading

Loria et al. devised an ex vivo organotypic-primary co-culture model of PD in which the efficacy of α-syn transfer between different cell types was investigated. They cultured hippocampal slices from wildtype mice, which were then exposed to 1 µM of sonicated Alexa-488-labeled α-syn fibrils for 16 h. After rinsing the slices, fluorescent ROSAmT/mG primary astrocytes were apically added to the slices. The authors noted that the exogenous astrocytes integrated and extended processes into the slice culture by three days. α-syn fibrils were also observed in the ROSAmT/mG astrocytes by 3–6 days. The opposite experiment was also performed, in which α-syn fibrils were added to primary ROSAmT/mG astrocyte culture for 16 h, and the astrocytes were then added on top of non-treated slice cultures. In this case, α-syn puncta were observed in endogenous slice culture astrocytes after 3–6 days, but not in neurons [65]. Therefore, the authors concluded that α-syn is transferred efficiently between astrocytes, but not from astrocytes to neurons. Interestingly, in the experiment performed by Elfarrash et al., exogenous α-syn fibrils were initially pipetted onto slice cultures rather than microinjected directly into the hippocampus. In this case, no α-syn fibrils were detected in neurons within either the DG, CA1, or CA3. The authors speculated that this may be due to the astrocytic “cap” that often forms on the apical surface of organotypic slice culture, which likely shielded underlying neurons from the exogenous α-syn [64].

### 4.5.α- Syn Strains

Organotypic slice cultures have also been used to study the seeding of different polymorphs or strains of α-syn fibrils. Shrivastava et al. cultured hippocampal slices from WT mice and added five different types of α-syn fibrils to the cultures. These different polymorphs were termed fibrils, ribbons, fibrils-91, fibrils-65, and fibrils-110. Some polymorphs were found to bind better to the surface of slice cultures than others, with fibrils-91 having the greatest affinity. The rate at which processes became positive for p129 α-syn within the slices also varied depending on the polymorph. Fibrils-91 induced immunopositive processes after four days, but immunopositive processes were not detected in slices treated with fibrils or ribbons until day 7. The number of α-syn aggregates seen in neuronal cell bodies also varied by polymorph, with many aggregates induced by fibrils-91 and fewer by ribbons. The type of polymorph added also determined which cell types were positive for p129 α-syn. Only ribbons led to p129 α-syn being detected in oligodendrocytes [66].

## 5. Slice Culture to Study Prion-Like Mechanisms in Amyotrophic Lateral Sclerosis

### 5.1. Induction of TDP-43 Aggregation

Several groups have produced either brain or spinal cord slice culture models for the study of ALS by adding compounds to induce cellular stress and TDP-43 pathology. For instance, Leggett et al. added tunicamycin, a protein N-glycosylation inhibitor, to cortical mouse brain cultures. This led to cytoplasmic TDP-43 inclusions and a reduction in nuclear TDP-43 immunoreactivity, which is a phenomenon known to occur in ALS [67]. Similarly, Alfroz and colleagues treated cortico-hippocampal slice cultures from wildtype mice with sodium arsenite to induce oxidative stress. Again, this compound led to oligomerization of TDP-43 in the cytoplasm. Interestingly, they observed that nuclear TDP-43 naturally exists in an oligomeric state and is structurally distinct from oligomers formed in the cytoplasm due to oxidative stress [68].

Spinal cord cultures have also been used to study lower motor neuron pathology in ALS. Ayala et al. treated rat lumbar spinal cord slices with the glutamate transport inhibitor D,L-threo-hydroxyaspartate (THA) for three weeks to induce excitotoxicity. TDP-43 inclusions formed in the cytoplasm of both neurons and astrocytes and there was a selective loss of neurons within the ventral horn region [69].

### 5.2. Addition of Exogenous TDP-43

Although many researchers have observed TDP-43 aggregation in slice culture after adding stress-inducing compounds such as tunicamycin, there have not been many ex vivo studies investigating prion-like spreading of TDP-43. However, exogenous TDP-43 can be added to organotypic slice cultures to facilitate understanding of its down-stream pathological effects in ALS or FTD. Specifically, Leal-Lasarte investigated how microglia may be involved in TDP-43-induced toxicity. The TDP-43 aggregates were added to spinal cord slice cultures of wildtype mice. Subsequently, TDP-43-positive inclusions were observed in microglia, which was accompanied by increased IL-18 secretion, activation of caspase-3, and upregulation of MAPK/MAK/MRK overlapping kinase (MOK) [70].

### 5.3. Application of Exogenous SOD1 to Promote Aggregation

Prion-like spreading and strain properties of SOD1 have also been studied in organotypic slice culture. Ayers et al. cultured spinal cord slices from transgenic mice expressing a G85R SOD1-YFP fusion protein and exposed them to spinal cord homogenates from aged G85R SOD1 mice that had been serially inoculated with either WT SOD1 fibrils or G93A, G37R, or L126Z SOD1 mouse brain homogenate. In all cases, SOD1 inclusions formed in slice culture and increased in number throughout the culturing period. Interestingly, each type of brain homogenate yielded different morphologies of SOD1 aggregates. As with in vivo inoculation, G93A homogenate led to round inclusions, while wildtype fibrils resulted in fibrillar aggregates. Although G37R and L126Z homogenates led to round inclusions in vivo, fibrillar aggregates were observed in slice culture [9]. Thus, there may be subtle environmental differences in slice culture compared to in vivo that influence SOD1 aggregation. The same group also added spinal cord homogenates from patients with sporadic or familial ALS or from non-ALS controls to mouse spinal cord cultures. Intriguingly, only brain homogenates from patients with the A4V SOD1 mutation led to the formation of SOD1 inclusions in culture. Therefore, the authors concluded that this particular mutation confers infectious prion-like properties to SOD1 that are not observed in sporadic cases of ALS [71].

### 5.4. Drug Screening

Organotypic slice culture may provide a useful platform for the screening of drugs that alter the aggregation propensity of SOD1 and its associated pathology. Rasouli et al. investigated how the addition of various acyl groups to SOD1 in vitro affects the ability of resulting fibrils to induce aggregation in slice culture. The addition of different acyl groups led to differing propensities of SOD1 to form either ThT-positive fibrils, ThT-negative fibrils, or amorphous aggregates. For instance, pyromellitylation was found to be the most effective at inhibiting SOD1 fibrillization and led to the production of ThT-negative fibrils. When the resulting SOD1 aggregates were added to spinal cord cultures from transgenic mice expressing G85R SOD1-YFP, only ThT-positive fibrils led to the formation of SOD1-YFP inclusions [72].

## 6. Slice Culture to Study Prion-Like Mechanisms in Huntington’s Disease

### 6.1. Induction of Human mHtt Aggregation

Organotypic slice culture has been a useful tool for the study of mHtt aggregation and its downstream pathological effects. One of the most widely used in vivo HD models is the transgenic R6/2 mouse, which expresses mHtt with ~120 CAG [73]. Smith et al. cultured hippocampal R6/2 slices and observed nuclear mHtt aggregates in CA1 by two weeks in culture and in CA3 and the dentate gyrus by three weeks, which mirrors the progression of aggregate development in vivo [73]. Reinhart et al. produced another ex vivo HD model in which a gene gun was used to deliver a mutant huntingtin construct containing 73 CAG repeats to coronal brain slices from Sprague-Dawley rats. After 3–5 days, significant degeneration of medium spiny neurons in the striatum and the formation of perinuclear mHtt aggregates was observed [74].

### 6.2. Prion-Like Spreading of mHtt

Organotypic slice culture has also been used to study the prion-like spread of mHtt. Pecho-Vrieseling et al. injected GFP-expressing human embryonic stem cell-derived neurons into either the cortex, striatum, or hippocampus of coronal slice cultures from R6/2 mice. It was observed that the human neurons were able to integrate into the slice culture and form synaptic connections with slice culture neurons. As early as two weeks, human neurons added to the cortex or striatum developed mHtt aggregates. In the cortex, a second wave of aggregate formation occurred after 6–8 weeks of culture. Other Huntington’s disease features were recapitulated in the human neurons, including loss of synaptophysin expression in presynaptic terminals and a reduction of primary and secondary neurites. Furthermore, addition of botulinum toxins, which inhibit neurotransmitter release and synaptic transmission, prevented the spread of mHtt to human neurons [75]. This suggests that the aggregates were being spread transynaptically. The authors performed a second experiment in which the R6/2 cortex was cultured with wildtype striatum, or wildtype cortex was cultured with R6/2 striatum. By four weeks of culture, it was observed that mHtt had spread from the R6/2 cortex to medium spiny neurons in the wildtype striatum. However, the opposite movement of mHtt from R6/2 striatum to wildtype cortex did not occur, indicating a directionality to the prion-like spreading [75].

### 6.3. Manipulation of Microglia

The ex vivo HD model developed by Reinhart et al. has been used to explore the contribution of inflammatory cytokines and microglia to HD pathology. Khoshnan et al. found that when slices were transfected with a level of mHtt that was normally insufficient to induce neurodegeneration, co-transfection with IL-34 led to degeneration of MSNs. However, when slices were transfected with a superthreshold quantity of mHtt, loss of MSNs and aggregation of mHtt could be prevented by addition of PLX3397 to cultures, which blocks the activation and proliferation of microglia [76].

### 6.4. Drug Screening

Ex vivo models of HD have also been useful for testing potential Huntington’s drug candidates. Murphy and Messer developed a HD model in which cortico-striatal slices from BALB/c ByGr mice were exposed to malonate to induce mitochondrial stress. The slices were also biolistically transfected with plasmids expressing either mutant (72 CAG) or normal (25 CAG) Htt. Malonate-treated slices transfected with mHtt plasmids exhibited significantly greater cell death compared to slices transfected with the non-mutant Htt plasmid. Cell death in these slices was significantly prevented when slice cultures were also co-transfected with a plasmid expressing an N-terminal Htt intrabody that had previously been shown to prevent mHtt aggregation in cell culture [77,78]. Using their hippocampal R6/2 slice culture model, Smith et al. found that the anti-amyloid compounds congo red and chrysamine G effectively inhibited mHtt aggregate formation. Creatine was also found to be effective [73]. Additionally, Reinhart et al. observed that the histone deacetylase inhibitor suberoylanilide hydroxamic acid and the adenosine 2A receptor antagonist istradefylline provided strong neuroprotection against MSN death in their slice culture model [74].

## 7. Advantages and Disadvantages of Organotypic Slice Culture for Neurodegenerative Disease Research

Depending on the scientific question being posed, slice culture models can provide an optimal compromise between in vivo and in vitro models (Table 1).

### 7.1. Economic and Ethical Advantages of Slice Culture

Organotypic slice culture offers many advantages over in vivo and cell culture experiments for studying protein misfolding diseases. Because many genetically identical slices can be obtained from a single mouse brain, significantly fewer animals are required for experimental replicates. This, coupled with the fact that slice culture prevents the need for animals to suffer from disease, makes it a more ethical alternative to in vivo studies. For these same reasons, slice culture is far less expensive than in vivo studies. The accelerated accumulation of PrP^Sc^ and pathology in POSCA and Aβ and tau in some AD slice culture models [37] also means that ex vivo experiments often take less time than in vivo studies.

### 7.2. Seeded Aggregation

By taking advantage of the prion-like nature of Aβ, tau, α-syn, TDP-43, SOD1, and mHtt, organotypic slice culture models of the corresponding neurodegenerative diseases have been significantly enhanced (Table 2). For instance, while many transgenic mouse lines such as tgCRND8 and APPPS1 have been used to model and study the development of Aβ aggregation and associated AD pathology, nontreated organotypic slice cultures from these models fail to develop Aβ aggregates by the time that plaque development occurs in vivo [35,36,37]. However, aggregation can be induced in slice culture through addition of exogenous Aβ aggregates [36,40]. Such seeded aggregation has also been successful through addition of tau fibrils to P301S slices [49], α-syn fibrils to WT mouse slices [64], and SOD1 fibrils to spinal cord cultures from transgenic mice expressing mutant G85R SOD1 [71].

### 7.3. Cellular Manipulation

In contrast to live animal studies, organotypic slice culture provides the unique opportunity to systematically manipulate specific populations of cells within their in vivo-like environment so that their contribution to disease pathology can be better understood. In many experiments, the role of microglia in clearance was investigated by their ablation with clodronate or PLX3397 [40,76]. Other groups have demonstrated that primary astrocytes and even embryonic stem cell-derived neurons can be functionally integrated into slice culture so that their roles both in releasing and accumulating protein aggregates could be elucidated [31,65,75].

### 7.4. Genetic Manipulation

Slice culture is also particularly amenable to genetic manipulation. Croft et al. have demonstrated the potential of rAAVs to induce stable expression of wildtype or mutant proteins in all or specific subsets of cells within organotypic cultures [62]. The use of gene guns has allowed for expression of mHtt in rat slice culture [74]. Additionally, slice culture is also uniquely suited to studying the effects of genetic mutations that would normally be lethal in living animals beyond the perinatal period.

### 7.5. Prion-Like Spread

Slice culture has proven to be an excellent platform for studying prion-like spread in proteinopathies, offering advantages over both in vitro and in vivo systems. While slice culture is more expensive and technically challenging to prepare than in vitro cell culture, usually only one cell type is used in simple cell culture, whereas all relevant neuronal and glial cell populations are present in organotypic slices with their connections and cytoarchitecture largely intact. This allows for investigation into the mechanisms of prion-like spread between different cell types and regions. In addition, within the prion field specifically, only a few immortalized cell lines are able to propagate prion infection, limiting the pathogenic questions that can be asked from these models.

Arguably, organotypic slice culture offers greater temporal resolution than in vivo models for tracking prionotypic spread of protein aggregates between cells and brain regions over short time periods. The propagation of α-syn aggregates both between subfields of the hippocampus and plexes of guinea pig ileum has been mapped over a period of days in slice culture [61,64]. Such precise timing of spread would be much more difficult in vivo and/or would require significantly more animals. However, because slice cultures thin over time and can only be maintained for about 3 months, in vivo studies would be more appropriate for longer term investigation of prion-like spread and an understanding of downstream pathology that is slower to develop. Of course, in vivo studies would also be the only option for studying the trafficking of protein aggregates from the periphery to the central nervous system. Additionally, it important to consider the trauma and inflammation that inevitably arise in organotypic culture due to the slicing of tissue. This trauma is known to generate a layer of astrocytic scar tissue on the apical surface of slices, which may alter the mechanisms by which proteins aggregate and spread compared to in vivo [64].

### 7.6. Drug Screening

The lack of a blood–brain barrier in organotypic slices makes them ideal for drug screening. Their open environment also allows for live imaging and real-time observation of pathology, providing faster readouts of drug effects than would be possible in vivo. Many authors have demonstrated the utility of slice culture to test drugs that inhibit aggregation or spread of Aβ [45,46], tau [48,79], α-syn [63], SOD1 [72], and mHtt [73,76,77]. Electrophysiological readouts, such as changes in LTP, are also possible in this ex vivo system [46]. An obvious advantage of in vivo models, however, is the possibility for behavioural tests of drug function, which may be more clinically relevant.

### 7.7. Strain Features

Several studies have demonstrated that organotypic slice cultures are capable of recapitulating strain features of protein misfolding diseases. Different sources of exogenous misfolded protein have led to variation in aggregate morphology for both Aβ and SOD1 in slice culture models of AD and ALS respectively [36,71]. Additionally, different polymorphs of α-syn fibrils were shown to induce phosphorylation of α-syn at different rates in slice culture and had variable effects on different cell types [66].

## 8. Suggestions for Future Investigation

### 8.1. Strain Tropism

Although several groups have shown that strain features are recapitulated in slice cultures, there has been a lack of investigation into the mechanisms by which strains lead to divergent pathology or why different strains preferentially target different regions or cell populations. Organotypic slice culture is an ideal system for exploring such topics. It is possible to expose a single region of a slice to an exogenous protein, or one could instead choose to bathe the entire slice in the prion-like agent, allowing all regions and cell populations to be exposed equally. The first type of experiment would enhance understanding of how connectivity between regions within a slice contributes to strain tropism, whereas the second type of experiment would facilitate understanding of cell susceptibility to different strains. One can also compare slices prepared from different brain regions. Comparison of differing slice orientations (i.e., coronal vs. sagittal) would also enhance understanding of how specific neuronal circuits contribute to spread between regions.

### 8.2. Slice Age

Due to the low viability of slices from adult mice and rats, slice culture has usually been prepared from neonatal animals. However, age is the greatest risk factor for developing most neurodegenerative diseases [80]. Therefore, adult slices would provide a more relevant environment for studying these proteinopathies. This point is exemplified by Hellwig et al., who show that microglia from neonatal but not adult mice prevent Aβ deposition in slice culture [40]. Similar findings have been reported by Daria et al., who observed that coculturing of aged APPPS1 plaque-containing slices with neonatal WT slices led to the mobilization of old microglia to Aβ plaques and clearance of plaque halos [81]. Some progress has been made in identifying factors that increase the viability of adult tissue. Specifically, Humpel suggests culturing thinner slices (100–150 µm) from adults [82]. Other authors have found that adult mouse brain slices have greater viability with serum-free medium [83,84]. Treatment of slices with brain-derived neurotrophic factor has also proven beneficial [85]. Another possibility is to add compounds to neonatal slice culture to transform their epigenetics to resemble more elderly tissue. This has been accomplished in pluripotent stem cell models by adding progerin, a truncated form of the lamin-A protein, which is implicated in the accelerated aging syndrome Hutchinson-Gilford progeria [86].

### 8.3. Transgenic vs. Wildtype Slices

Another consideration is that many of the aforementioned slice culture models are prepared from transgenic mice that express mutant versions of the proteins being studied. Many of these transgenic mice, such as the 5xFAD, APP23, and APPPS1 mice used to study AD, and the R6/2 mice used to study HD, develop aggregates in vivo after weeks to months [36,40,73]. Therefore, it is probably more accurate to say that prion-like aggregation and spreading is being accelerated rather than induced by addition of exogenous protein fibrils to slice cultures prepared from these mouse models. Furthermore, most of these models develop protein aggregates with specific morphologies and tropism, meaning that they have an underlying default strain. Therefore, the pathology elicited by adding prion-like seeds is likely influenced to some extent by the default strain of the transgenic mouse model. For this reason, and the fact that most neurodegenerative diseases are sporadic and not related to a genetic mutation, it would be useful to study prion-like spreading in slice culture models expressing non-mutant human proteins. Table 3 provides a summary of the various transgenic and wildtype animals from which slice culture has been prepared to study proteinopathies.

### 8.4. Comorbidity of Protein Misfolding Diseases

It is possible that the phenotypic diversity seen in neurodegenerative diseases is due to alternate strains or pathological conformations of the culprit proteins, but variation between patients may also arise from the coexistence of two or more proteinopathies. For instance, although AD is primarily associated with Aβ and tau aggregates, approximately 50% of cases also contain α-syn pathology [87]. Moreover, tau tangle pathology is found to coexist not only with Aβ in AD, but also with PrP^Sc^ in prion diseases, TDP-43 in frontotemporal dementia, and α-syn in Lewy body dementia [88]. For this reason, slice culture could be used to better understand how prion-like proteins influence the aggregation and spread of each other, and the resulting pathology. The method of using rAAVs demonstrated by Croft et al. shows promise for customizing slice cultures to express one or a combination of wildtype or mutant prion-like proteins discussed in this review [62].

## 9. Conclusion

Organotypic slice culture offers a valuable, cost-effective intermediate between in vitro and in vivo experiments for investigating neurodegenerative disease. It is a versatile ex vivo model that can be used to enhance understanding of disease mechanisms in protein misfolding neurodegenerative disorders. This system retains many features of the in vivo environment but is open to drug manipulation and live imaging of disease processes. Compared to in vivo models, it is much easier to add exogenous protein aggregates to slice culture to induce prion-like aggregation, spreading, and downstream pathology. This has been demonstrated in slice culture with several prion-like proteins, including Aβ, tau, α-syn, TDP-43, SOD1, and mHtt. In conclusion, capitalizing on the prionotypic nature of these proteins in slice culture models offers unique opportunities to study and model strain characteristics in neurodegenerative diseases, which likely underlie much of the phenotypic heterogeneity seen between patients.

## Figures and Tables

**Figure 1 biomolecules-10-01079-f001:**
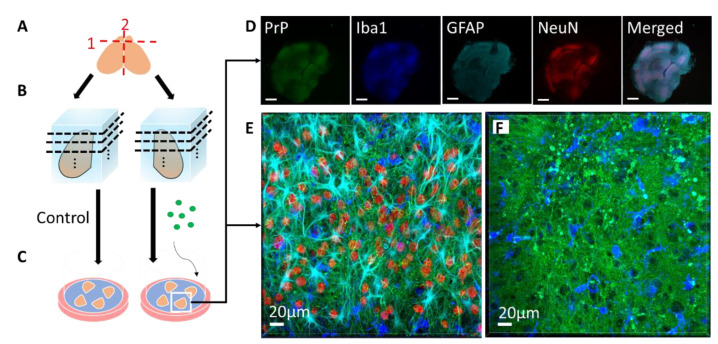
Schematic of coronal slice culture preparation, infection, and confocal imaging. (**A**) The brain is removed from 8- or 9-day-old mouse pups, the olfactory bulbs are removed, and the brain is hemisectioned sagitally. (**B**) The hemispheres are embedded in low-melting-point agarose gel and sliced into 275 µm-thick coronal sections with a vibratome. About 40 slices anterior-posterior are obtained from each mouse, with 4 slices cultured per well. Each well is considered a “region”, from 1 (anterior) to 10 (posterior), each representing 1100 µm. (**C**) The slices are then placed on Milicell cell culture inserts with the culture media below the membrane and can be cultured for as long as three months. During the culture period, genetically identical, location-matched slices can be used as controls or subjected to different treatment conditions in the absence of the blood–brain barrier, such as infection with different prion strains or treatment with different drugs. At various time points, the slices can be immunostained for confocal imaging. (**D**) Low magnification confocal image of an RML-infected slice that was cultured for 56 days post-infection. PrP (Saf83) is shown in green, microglia (Iba1) are shown in blue, reactive astrocytes (GFAP) are shown in cyan, neuronal nuclei (NeuN) are shown in red. As seen in the NeuN channel, the boundaries of distinct brain regions are discernable (scale = 500 µm). (**E**) High magnification confocal image of a neocortical region from the RML-infected slice in E, showing the array of cell types and architecture. (**F**) High magnification confocal image of an RML-infected slice showing PrP aggregates (green) and activated microglia (blue).

**Figure 2 biomolecules-10-01079-f002:**
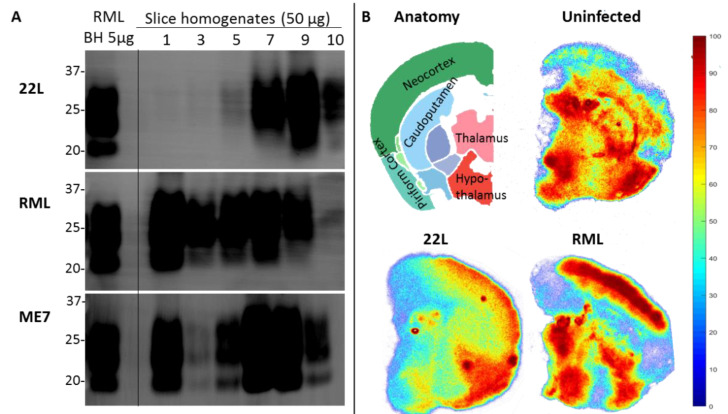
Coronally sliced whole brain prion organotypic slice culture assay (POSCA) can propagate three rodent-adapted scrapie strains. (**A**) Immunoblots of proteinase K (PK)-digested PrP from slice homogenates of anterior (1) through posterior (10) cultured brain regions. Slices were infected with 22L, RML, or ME7 strain of rodent-adapted scrapie, then harvested at day 56 post-infection. All strains can be propagated in this system. Whether the different levels of PK-resistant PrP are strain-specific is the question of ongoing experiments. POSCA was prepared from C57Bl6 mice for all strains, with the same baseline PrP^C^ levels. An amount of 50 µg total protein before PK digestion was loaded. (**B**) Anatomy of a slice from region 6 and heat maps for total PrP (PrP^C^ plus PrP^Sc^) of tga20* POSCA uninfected or infected with 22L and RML, showing different distributions of total PrP. Heat map values indicate relative amounts (percentiles of total PrP) within a slice, not absolute values of PrP, so it is the pattern of total PrP distribution, not the intensity of signal, that can be compared between slices. Unlike immunoblots which only provide total levels of a protein from a homogenate, slice culture allows analysis of regional variation within a single slice. * tga20 mice express wildtype mouse PrP at 6× fold higher levels and were the original mouse line used for the first POSCA experiments. PrP antibodies: Sha31 (**A**), SAF83 (**B**).

**Table 1 biomolecules-10-01079-t001:** Advantages and disadvantages of organotypic slice culture vs. in vivo or cell culture studies.

	In Vivo	Slice Culture	Primary Culture	Simple Cell Culture
Cost	+ Housing for duration of experiment, high animal numbers	++ Breeding costs, fewer animals	++ Breeding or animal purchase costs, fewer animals	+++
Time	+ Months	++ Weeks	++ Weeks	+++ Days
Ethics of animal use	+ High animal numbers, induction of symptomatic disease	++ Fewer animals, no need to induce disease	++ Fewer animals, no need to induce disease	+++ No animals required
Technical difficulty	+ Inoculation, clinical assessment	++ Fast, accurate dissection, orientation for slicing	++ Fast, accurate dissection	+++ Sterile technique
Cell types, cyto-architecture	+++ All cell types, connections intact	++ All cell types, some connectivity disrupted	- Often one cell type No cytoarchitecture	- Often one cell type No cytoarchitecture
Genetics	++ Many models	+++ Can use any mouse models PLUS transgenic models that are lethal beyond the perinatal period, can transfect with recombinant adeno-associated viruses, can have many genetically identical slices from same mouse	+ Many genetically identical cell populations can be obtained from a single animal	- Immortalized cell lines are often cancer-based and can be unstable genetically
Real time monitoring	++ Closed system, monitor behavioural/clinical phenotype	+++ Open system, more amenable to live-cell imaging, no phenotype	+++ Open system amenable to live-cell imaging, no phenotype	+++ Open system amenable to live-cell imaging, no phenotype
Animal age	+++ Any age	++ Most viable cultures are from neonatal animals	+ Tissue must be taken from prenatal or neonatal animals	n/a
Vasculature	+++ Intact, blood–brain barrier present	++ No blood–brain barrier—better for drug testing	- None	n/a

+++ most advantageous; ++ less advantageous; + least advantageous. n/a indicates feature is not applicable in that model.

**Table 2 biomolecules-10-01079-t002:** Overview of organotypic slice culture studies which have used the prion-like features of aggregate-prone proteins from other neurodegenerative diseases.

Prion Feature	Amyloid Beta	Tau	α-Synuclein	TDP-43	SOD1	Huntingtin
Seeded aggregation	[36,40,41]	[49]	[63,64]		[71]	
Prion-like spreading		[50,51]	[64,65]			[75]
Pathology induced by seeding	[36,47]		[63]	[70]		[75]
Strains	[36]		[66]		[71]	
Drug screening (to prevent aggregation/spread)	[45,46]	[48]	[63]		[72]	[73,76,77]

**Table 3 biomolecules-10-01079-t003:** Summary of in vivo and ex vivo experiments for prion-like protein aggregates, indicating type of mouse model used for culture and whether application of a prion-like seeding agent facilitated aggregation or pathology.

Ref.	Animal Model	In Vivo Pathology	Unseeded Slice Pathology	Prion-Like Seeding Agent	Seeded Slice Pathology
**Amyloid beta**					
[35]	CRND8 mice: 5x expression of human APP with Swedish and Indiana mutations [89]	ThS-positive plaques by 3 months Dense-cored plaques by 5 months [89]	No ThS-positive Aβ aggregates by 9 weeks of culture Accumulation of intracellular Aβ in axonal swellings	Synthetic Aβ	No ThS-positive Aβ aggregates by 9 weeks of culture
[36]	APPPS1 mice: 3x expression of human APP with Swedish mutation human PS1 with L166P mutation [90]	Aβ plaques by at 6 weeks [90]	No evidence of Aβ aggregates by 10 weeks	Single treatment with aged APPPS1 or APP23 brain homogenate and continuous supplementation with synthetic Aβ	Extensive Aβ aggregates after 1 week Aggregate morphology dependent on source of brain homogenate (APPPS1 or APP23)
	APP23 mice: 7x expression of human APP with Swedish mutation [91]	Aβ plaques by 6 months [91]			
	WT mice	N/A			
	APP-null mice	N/A			
[40]	WT mice	N/A	No evidence of Aβ aggregates	Treatment with Clodronate to remove microglia and 4 treatments with synthetic Aβ_42_	ThS-positive Aβ aggregates after two weeks
[41]	WT mice	N/A	No evidence of Aβ aggregates	Treatment with clodronate to remove microglia and addition of Aβ_42_ oligomer solution	Increased ThT fluorescence after 1 week (suggests Aβ aggregate formation)
**Amyloid beta and tau**					
[37]	3xTg-AD mice: Express human APP with Swedish mutation, human mutant P301L tau, and human PSEN1 with M146V mutation [92]	Extracellular Aβ deposits by 6 months Aggregates of hyperphosphor-ylated tau by 12–15 months [92]	No evidence of Aβ or tau aggregates after 28 days of culture Accelerated accumulation of Aβ42 and phosphorylated tau compared to in vivo	N/A	N/A
**Tau**					
[48]	Tg Mice expressing human 4R tau with the ΔK280 FTD mutation [93]	Hyperphosphor-ylation and aggregation of tau by 5–10 months [93]	ThS-positive cell bodies by 20 days	N/A	N/A
[49]	PS19 tau mice: Express human 1N4R Tau with P301S mutation [94]	PHF1-positive neuronal staining by 6 months in hippocampus, amygdala and spinal cord [94]	No evidence of tau aggregation after 13 days of culturing	Recombinant tau fibrils added on days 3 and 6 of culturing	Tau aggregation in hippocampal CA1 neurons 10 days after first seeding
	WT mice	N/A			No evidence of tau aggregation 10 days after first seeding
**α-synuclein**					
[63]	WT rats	N/A	N/A	Treated on day 13/14 with monomeric α-syn, fibrillar α-syn, or a mixture of both exogenous forms	α-syn monomers did not cause cell death Mixture of α-syn monomers and fibrils significantly more toxic than α-syn fibrils alone
[64]	WT mice	N/A	N/A	α-syn fibrils microinjected into the dentate gyrus	α-syn aggregates appeared in the dentate gyrus after 3 days and in CA1 and CA3 by 3–5 days (no evidence of aggregates when monomeric α-syn injected)
	SNCA knockout mice				No evidence of α-syn aggregates
[65]	WT mice	N/A	N/A	Primary ROSAmT/mG astrocytes were incubated with Alexa-488-labeled α-syn fibrils for 16 h. The astrocytes were then added on top of slice culture	α-syn inclusions observed in slice culture astrocytes but not neurons after 3–6 days
[66]	WT mice	N/A	N/A	α-syn fibrils of 5 different polymorphs were added to slice cultures	α-syn aggregates were observed after 4–7 days. Extent and rate of aggregation depended on fibril polymorph
**SOD1**					
[9]	Tg mice expressing human G85R mutant SOD1-YFP fusion protein [95]	Develop fluorescent SOD1 puncta in anterior horn of spinal cord by 9 months in cell bodies and neuropil [95]	(spinal cord culture) N/A	Treatment with spinal cord homogenates from paralyzed G85R SOD1:YFP mice that had been occulated with different “mouse-adapted” ALS strains^1^	SOD1 aggregates induced in spinal cord slices Morphology of aggregates were dependent on strain
				Treatment with spinal cord homogenates from sporadic or familial ALS patients	SOD1-YFP punctate inclusions by 7 days only when treated with A4V SOD1 mutation
[72]	Tg mice expressing human G85R mutant SOD1-YFP fusion protein [95]	Develop fluorescent SOD1 puncta in anterior horn of spinal cord by 9 months in both cell bodies and neuropil [95]	(spinal cord culture) N/A	WT SOD1 was modified with various acyl groups and aggregated in vitro. The resulting ThT-positive or negative fibrils were then added to slice culture.	After 1 month of culture, only treatment with ThT-positive fibrils led to the formation of SOD1-YFP inclusions
**Huntingtin**					
[73]	R6/2 mice: Express mHtt with ~115–150 CAG repeats [96]	Develop mHtt aggregates in CA1 of hippocampus by 3 weeks and in CA3 by 5 weeks [73]	mHtt aggregates are observed in CA1 of the hippocampus by 2 weeks and in CA3 and dentate gyrus by 3 weeks [73]	N/A	N/A
[75]	WT mice	N/A	Cortex or Striatum culture N/A	WT-R6/2 Cocultures: WT striatum with R6/2 cortex WT cortex with R6/2 striatum	mHtt aggregates spread from R6/2 cortex to MSNs in WT striatum by 4 weeks, but not from R6/2 striatum to WT cortex

All cultures are hippocampal unless otherwise indicated. ^1^ In the experiment by Ayers et al., (2016), G85R-SOD1-YFP mice were inoculated with homogenates from various mutant SOD1 mouse lines, including G93A, G37R, and L126Z, or with WT SOD1 fibrils. Spinal cord homogenates were then taken from these initially inoculated mice and passaged a second time into G85R-SOD1-YFP mice. Homogenates from these mice were then used to seed SOD1 aggregation in spinal cord slice cultures from G85R-SOD1-YFP mice.

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
