# Peer review of "POSCAbilities: The Application of the Prion Organotypic Slice Culture Assay to Neurodegenerative Disease Research"

_biomolecules, 2020, doi:10.3390/biom10071079_

Round 1
Reviewer 1 Report
The manuscript provides a detailed review on the use of organotypic slice cultures for the study of prion diseases and other CNS disorders displaying prion-like characteristics.
Despite presenting interesting and related to the topic information, I am afraid that additional information is presented, rendering the manuscript less reader friendly and beyond the desired length limit.
To improve this, I would suggest focusing on the organotypic slice culture models that can be used for the study of prion-like characteristics of other NDs. These may include studies referring to: a) seeded aggregation (representing prion or prion-like transmissibility), b) pathology spread within the culture, (including the identification of the involved cell types) and c) data supporting strain specific characteristics. I would also suggest presenting pharmacological studies in which the tested drug(s) affect(s) any of the before mentioned prion-like characteristics.
In this regard, I would also suggest amending the provided Table so as to include the prion-like specific characteristic(s) that is(are) replicated by each of the presented studies (e.g. seeded aggregation, strain differentiation, pathogenetic mechanism -spread of aggregation, cell-type dependent spread of aggregation, underlying molecular mechanism, pharmacological intervention).
I believe that a rearrangement of the presented information will significantly help to improve the manuscript. A potential outline could include i) prion diseases and other NDs-main aggregated proteins per disease, ii) evidence of prion-like characteristics in the presented NDs, iii) organotypic slice culture for the study of prion diseases, iv) organotypic slice cultures available for the study of NDs, v) studies on organotypic slice cultures of other NDs in regard to their prion-characteristic properties, vi) comparison of POSCA with other organotypic slice cultures, vii) advantages and disadvantages of organotypic slice cultures.
Regarding the references:
- please check whether reference 37 has been correctly cited (all author names included).
- The following References may also be included in the corresponds sections of the manuscript
Croft CL, Wade MA, Kurbatskaya K, Mastrandreas P, Hughes MM, Phillips EC, Pooler AM, Perkinton MS, Hanger DP, Noble W. Membrane association and release of wild-type and pathological tau from organotypic brain slice cultures. Cell death & disease. 2017 Mar;8(3):e2671-.
Croft CL, Kurbatskaya K, Hanger DP, Noble W. Inhibition of glycogen synthase kinase-3 by BTA-EG 4 reduces tau abnormalities in an organotypic brain slice culture model of Alzheimer’s disease. Scientific reports. 2017 Aug 7;7(1):1-1.
Daria A, Colombo A, Llovera G, Hampel H, Willem M, Liesz A, Haass C, Tahirovic S. Young microglia restore amyloid plaque clearance of aged microglia. The EMBO journal. 2017 Mar 1;36(5):583-603.
Jang S, Kim H, Kim HJ, Lee SK, Kim EW, Namkoong K, Kim E. Long-term culture of organotypic hippocampal slice from old 3xTg-AD mouse: An ex vivo model of Alzheimer’s disease. Psychiatry investigation. 2018 Feb;15(2):205.
Author Response
I have included the itemized responses to all reviewers in the attachment, as some of their suggestions help address some of the comments from other reviewers as well.

Reviewer 2 Report
This is a comprehensive review of applying organotypic slice culture to investigate neurodegenerative disease. The manuscript is well written and presents a nice overview of the current status of this field. A few areas can be further improved to help readers who are interested in this approach.
- Figure 1 appears to be the original study conducted in authors’ lab. Given that the result has not been published, more details, such as the exact position of slices 1 to 10, will be helpful for readers to understand the experiment. For figure 1A, it might be helpful to include the level of PrP (without PK and without prion infection) in each slice, which will show whether the difference in conversion in each slice is due to different prion strains or the differences in endogenous PrP in each slice.
- For figure 1B-G, there is no indication of the color scale for the heat map. Why the background staining in uninfected slice is so high? Are these after PK digestion? Are these slices from the same position? These results showed a difference in tropism of various prion strains in three dimensions. Fig 1A showed differences from front to back of the brain. Heat maps showed difference within a single slice (assuming they are from the same position). Authors may consider mentioning this point as an advantage for slice culture.
- The review covers all neurodegenerative diseases, but the abstract focuses more on prion disease. It might helpful to reorganize or rewrite the abstract to give an overview of the article.
- In the abstract, it states “In this review, we explore the advantages and disadvantages of this culture system…” But the advantage and disadvantage of the system is discussed in the context of prion disease. I suggest moving the advantage and disadvantage to the end to cover all the studies.
- I suggest changing the title of “Discussion”, to something like “advantages and disadvantages of the slice culture system”. All the points mentioned in the discussion could be discussed under this title.
Author Response

(The authors gave the same response as above.)

Reviewer 3 Report
The authors present a literature review on the results of the application of organotypic brain slice cultures (OBSC) to the study of neurodegenerative diseases (NDD). The methodology has been specifically introduced to study prion infection and pathogenesis, but then increasingly applied to study other NDD sharing a prion-like mechanism of cell-to-cell transmission.
Although focused on a methodology rather than a scientific problem, the review addresses a relevant topic and is generally well-written.
I have the following suggestions for improvements:
1) I would shorten or even eliminate all introductory paragraphs referring to each NDD. Also considering the overall length of the text, I think they distract the readers from the "main body" of the review, namely the critical revision of the scientific data collected to date on the application of OBSC to the study of prion-like NDD. Given that the recent literature includes several detailed reviews on this topic, the authors may refer to them the non-specialist readers.
2) There is also room for improvement in the discussion paragraph. This should include a critical summary/revision of the content of all previous paragraphs. Again, it would be essential to (critically) summarize the main scientific achievements (and failures as well) obtained through the application of OBSC and, in addition, provide an insight into the scientific issues that are expected to be solved/addressed through the application of this novel methodology.
3) The authors should consider adding a figure which schematically illustrates the main features of OBSC and, perhaps, their advantages over standard cell cultures.
Author Response

(The authors gave the same response as above.)

Round 2
Reviewer 3 Report
The authors have thoroughly considered my suggestions, I am happy with the revision.